# Autoimmune Polyendocrine Syndromes in the Pediatric Age

**DOI:** 10.3390/children10030588

**Published:** 2023-03-19

**Authors:** Roberto Paparella, Michela Menghi, Ginevra Micangeli, Lucia Leonardi, Giovanni Profeta, Francesca Tarani, Carla Petrella, Giampiero Ferraguti, Marco Fiore, Luigi Tarani

**Affiliations:** 1Department of Maternal Infantile and Urological Sciences, Sapienza University of Rome, Viale del Policlinico 155, 00161 Rome, Italy; 2Department of Experimental Medicine, Sapienza University of Rome, Viale del Policlinico 155, 00161 Rome, Italy; 3Institute of Biochemistry and Cell Biology, IBBC-CNR, 00185 Rome, Italy

**Keywords:** autoimmune, polyendocrinopathy, polyglandular, immunodeficiency, pediatrics, children, autoantibodies

## Abstract

Autoimmune polyendocrine syndromes (APSs) encompass a heterogeneous group of rare diseases characterized by autoimmune activity against two or more endocrine or non-endocrine organs. Three types of APSs are reported, including both monogenic and multifactorial, heterogeneous disorders. The aim of this manuscript is to present the main clinical and epidemiological characteristics of APS-1, APS-2, and IPEX syndrome in the pediatric age, describing the mechanisms of autoimmunity and the currently available treatments for these rare conditions.

## 1. Introduction

The autoimmune polyendocrine syndromes (APSs) are clusters of endocrine abnormalities characterized by sequential or concomitant deficiencies in the function of endocrine glands, with a combination of several autoimmune disorders and, in some cases, immunodeficiency [1]. Non-endocrine organs might also be affected; thus, the term “polyendocrine” (or “polyglandular”) could be itself misleading, since not all patients have multiple endocrine disorders, and many show non-endocrine autoimmune diseases. The clinical classification of APSs was proposed by Neufeld and Blizzard in 1980, and four main types were reported [2]. However, types 3 and 4 have been considered to be variants of APS-2. In fact, a recent nosological classification recognizes three main syndromes [3]: APS-1, APS-2, and immune dysregulation, polyendocrinopathy, enteropathy, X-linked (IPEX) syndrome (Figure 1). In particular, according to the 2022 update of the International Union of Immunological Societies, the phenotypical classification for human inborn errors of immunity, APS-1 and IPEX are labeled as syndromes with autoimmunity classified into diseases of immune dysregulation [4].

APS-1 and IPEX syndrome are monogenic disorders, caused by mutations in the autoimmune regulator (*AIRE*) and forkhead box protein P3 (*FOXP3*) genes, respectively [5,6]. APS-2 is a polygenic, complex genetic disorder associated with human leukocyte antigen (HLA) and non-HLA genes, representing the most frequent autoimmune polyglandular syndrome [3]. Based on the complexity of these diseases, the role of the pediatrician is crucial in the appropriate identification of the clinical picture and in supporting the family during the continuative process that involves the management of these patients [7,8,9,10].

In this narrative paper, the clinical and epidemiological characteristics of APSs are described, reporting on their presentation in childhood and adolescence, and discussing as well on pathogenesis, autoimmunity, and therapeutic options (Table 1).

## 2. Autoimmune Polyendocrine Syndromes

### 2.1. APS-1

Additionally known as autoimmune polyendocrinopathy–candidiasis–ectodermal-dystrophy (APECED), APS-1 is diagnosed by the presence of two of the following three conditions: chronic mucocutaneous candidiasis (CMC), hypoparathyroidism (HP), and Addison disease (AD). Patients generally present during early childhood with CMC, followed by HP and then by AD. Simultaneous presentation is exceptional. 

#### 2.1.1. Epidemiology

APS-1 usually manifests in infancy or early childhood, at the age of 3–5, or in early adolescence, with a male-to-female ratio of 3:4. It is a rare disorder, with an estimated incidence of < 1:100,000/year [11]; however, in some geographical areas, among genetically isolated ethnic groups (such as Finnish, Sardinian, and Iranian Jewish populations), it is relatively more frequent [12,13,14].

#### 2.1.2. Pathogenesis and Autoimmunity

APS-1 is an autosomal recessive, monogenic disorder that results from a mutation in the *AIRE* gene on chromosome 21q22.3 [15]. The transcription factor encoded by the *AIRE* gene has a crucial role in immune tolerance, controlling the expression of many thymic proteins. The decreased expression of the transcription factor subsequent to *AIRE* mutations leads to a reduced presentation of self-antigens by medullary thymic epithelial cells and dendritic cells to developing T-lymphocytes, eventually determining the autoimmune destruction of target organs by altering the immunological tolerance [16,17]. In addition, autoantibodies against type 1 interferons, especially antibodies to omega-interferon (IFN-ω-Abs) and alpha2-interferon (IFN-α2-Abs), are highly specific markers for APS-1 [18]. Assays for the measurement of IFN-ω-Abs and IFN-α2-Abs may be used as a convenient and rapid method for the selection of patients for *AIRE* gene analysis, ruling out the diagnosis in those who do not have IFN-ω-Abs and IFN-α2-Abs [19,20].

#### 2.1.3. Clinical Features

CMC is most often the first manifestation of APS-1, with 70% of cases occurring in children younger than 5 years of age [21]. Any child with refractory CMC should be evaluated for coexisting endocrine disorders, as up to almost 50% of children might develop the other typical components of APS-1 over time [22,23]. This recurring fungal infection frequently affects the nails, skin, tongue, and mucous membranes, due to the selective immunological deficiency of a normal T-cell-mediated response; the systemic infection generally associated with severe immunodeficiency is absent, given the normal B-cell response with anticandidal antibodies preventing the development of systemic candidiasis [24].

HP is the most frequent and, typically, the first endocrine component to present. Furthermore, 90% of cases occur after 3 years of age, after the presentation of CMC and usually preceding AD onset [21,25]. Clinical features of chronic HP include paresthesias, neuro-muscular hyperexcitability, hypotension, malabsorption, and steatorrhea [26]. Symptoms of hypocalcemia may be vague and non-specific for a long time. Hypocalcemia may be additionally masked in the presence of AD due to volume contraction and increased renal tubular reabsorption of calcium. Moreover, the parathyroids seem to be somehow preserved in APS-1 patients when AD occurs as the first disease component since further autoimmunity against the parathyroids is likely down-regulated due to the steroid replacement therapy [27,28]. Biochemically, patients display hypocalcemia and hyperphosphatemia with normal or low parathyroid hormone (PTH) levels. An autoimmune HP is very likely if associated with the presence of antibodies against parathyroid autoantigens (calcium-sensing receptor, CaSR; NACHT leucine-rich-repeat protein 5, NALP5), other autoimmune disorders, or organ-specific antibodies [22].

AD most commonly appears after CMC and HP, in 90% of cases after 6 years of age [21]. The pathophysiologic mechanism underlying the clinical manifestations of AD is the deficiency of glucocorticoids (leading to hypoglycemia, with associated nausea, vomiting, fatigue, and weakness) and mineralocorticoids (dizziness, headache, salt craving, dehydration, electrolyte abnormalities, hypotension, shock in severe cases) [29]. Hyperpigmentation of the skin and mucous membranes is caused by melanocyte-stimulating hormones derived from the adrenocorticotropic hormone (ACTH) precursor proopiomelanocortin, since the lack of negative feedback from cortisol, typical of primary adrenal insufficiency, leads to an augmented secretion of corticotropin-releasing hormone and ACTH [30]. In children previously diagnosed with AD, as well as in undiagnosed cases, clinical conditions may worsen until the stage of adrenal crisis, a metabolic emergency that may be triggered by infections, surgery, other stressful events, or by an interruption in replacement therapy [30]. Laboratory tests reveal low basal plasma cortisol, elevated ACTH, hyponatremia, hyperkalemia, low aldosterone, and elevated plasma renin activity. Autoantibodies can be present years before clinical or biochemical evidence of AD, directed against antigen targets represented by adrenal enzymes including P450scc (side-chain cleavage enzyme), P450c17 (17-alpha-hydroxylase), and P450c21 (21-hydroxylase) [31]. In particular, adrenal cortex/anti-21-hydroxylase autoantibodies might be detected in up to 48% of children showing the major components of APS-1 (HP with or without CMC) without overt AD. The presence of such antibodies indicates a high likelihood of the development of AD [32]. Considering that the time interval between the onset of CMC and HP might take up to five years in APS-1, and the subsequent involvement of adrenal glands may take up to ten years, once a child has been diagnosed with CMC and/or HP, screening should be done for the presence of the abovementioned adrenal autoantibodies, and rescreening, in the case of initial negative autoantibody tests, is mandatory.

Less common features include type 1 diabetes mellitus (T1DM), autoimmune thyroid disease (AITD), primary hypogonadism, vitiligo, pernicious anemia, alopecia, autoimmune hepatitis, and malabsorption syndrome [22]. Asplenism has also been reported by Friedman et al. in four out of nine patients with APS-1 [33]. However, pediatricians should be aware of the enlarging clinical spectrum of APS-1 in childhood, also including chronic inflammatory demyelinating polyneuropathy, chronic lung disease, and gastrointestinal dysfunction [34,35].

#### 2.1.4. Treatment

Since CMC can potentially affect any area along the gastrointestinal tract, it must be treated aggressively, monitored for recurrence, and strictly controlled to prevent cancer development [25]. Although a twice-daily PTH regimen has been found to provide stable calcium levels in children with HP, this treatment is not yet established as routine; thus, the mainstay of therapy is oral calcium and vitamin D supplementation [36]. Hormone replacement therapy in AD consists of daily hydrocortisone and fludrocortisone to treat the lack of glucocorticoids and mineralocorticoids, respectively. Every patient with AD, in particular children with their families, must be instructed on the rationale for replacement treatment and warned about the need for an emergency parenteral treatment or a modification in the case of intercurrent diseases or stress [37]. Patients should wear a medical bracelet or necklace bearing information such as their medical condition, prescribed medication, and contact details in case of emergency.

Other endocrinopathies of APS-1 are usually managed with hormonal replacement treatment. Initiation of levothyroxine before glucocorticoid replacement in APS-1 or APS-2 hypothyroid patients with AD can lead to the precipitation of adrenal crisis [38]. If asplenism is found, vaccinations against encapsulated bacteria (*Haemophilus influenzae* type b, *Neisseria meningitidis*, and *Streptococcus pneumoniae*) should be administered [39].

### 2.2. APS-2

As abovementioned, four main types of APSs were described by Neufeld and Blizzard in 1980. As reported by this classification: APS-2 refers to AD (always present) associated with AITD and/or T1DM; APS-3 is defined as AITD combined with other autoimmune diseases (excluding AD and/or HP); APS-4 includes all the clinical combinations which cannot be included in the previous groups [2]. According to a more recent approach, all the above combinations can be considered as APS-2 (thus leaving only APS-1 and APS-2, no longer recognizing APS-3 and APS-4 as independent entities) [3,40].

AD is the initial presentation in 50–70% of the patients and, in approximately 40–50% of cases with AD, additional autoimmune endocrinopathies occur [41]. Years to decades may pass between the onset of the first and second component disease of APS-2 [11].

#### 2.2.1. Epidemiology

APS-2 occurs more commonly than APS-1, typically in late childhood or early adulthood, with a male-to-female ratio of 1:3 and an estimated incidence of 1-2:100,000/year [41]. It is therefore a very rare disease in children, with a female preponderance. The disease is genetically complex; there is familial clustering, with parents, siblings, and offspring often affected by one or more component diseases [2]. 

#### 2.2.2. Pathogenesis and Autoimmunity

The underlying immunologic defect of APS-2 is still unknown, and the etiology remains to be determined. The inheritance is complex and polygenic, with the HLA system, a set of genes on chromosome 6 which is the human version of the major histocompatibility complex (MHC) encoding cell-surface proteins responsible for immune regulation, playing a predominant role [42]. The HLA gene complex is divided into three classes (I, II, and III). In particular, the class II alleles (HLA-DQ, HLA-DR, and to a lesser degree HLA-DP) encoding the MHC class II proteins expressed only on B-cells, activated T-cells, and antigen-presenting cells (monocytes, macrophages, and dendritic cells), represent the major factors of the underlying disorders of APS-2 [3].

Specific HLA alleles strongly influence which peptides are targeted and therefore which tissue will be affected by autoreactive T-cells, leading to organ-specific autoimmunity as a consequence of the loss of tolerance [1,3]. An increased prevalence of the HLA-DR3 and HLA-DR4 alleles has been found in patients with APS-2, conferring an increased risk for disease development [43]. The HLA-DR3/4-DQ2/8 haplotype, which can be found in 30% of patients with AD, specifically represents one of the highest-risk genotypes in APS-2 [3].

Polymorphisms in genes involved in other autoimmune disorders, including cytotoxic T-lymphocyte associated protein 4 (*CTLA-4*) and protein tyrosine phosphatase non-receptor type 22 (*PTPN22*) among others, have been identified as additional risk factors for APS-2 [44,45]. Based on the polymorphism, non-HLA genes in fact contribute to autoimmunity in APS-2, by either predisposing to a loss of tolerance lowering the threshold for autoimmunity onset, or conditioning the selection of the specific organ targeted by the autoimmune process. Despite the well-established role of the HLA gene complex in the pathogenesis, consideration must also be given to non-HLA genes as relevant factors that promote the co-occurrence of numerous autoimmune endocrinopathies in APS-2 [46].

Since there are no distinctive tests to identify patients with APS-2, children with a monoglandular autoimmune disease should undergo clinical and serological screening in order to intercept other glandular and/or non-endocrine autoimmune disorders. Patients with clinical and laboratory evidence of primary adrenal insufficiency should be tested for the presence of anti-21-hydroxylase autoantibodies, in order to confirm the autoimmune etiology of the disease. Once AD is diagnosed, screening for other endocrinopathies is recommended. In fact, subjects with AD have a 50% lifetime risk of additional autoimmune disease onset, thus making screening for other autoantibodies against organ-specific autoantigens (thyroid peroxidase [TPO], thyroglobulin [Tg] for AITD; insulin, glutamic acid decarboxylase-65 [GAD-65], islet antigen 2 [IA2], zinc transporter 8 [ZnT8] for T1DM; tissue transglutaminase for celiac disease; others) fundamental [3]. Therefore, in any patient with AD, the abovementioned screening is recommended and, if negative, should be repeated every 2–3 years [11]. Screening of APS is equally crucial in children with T1DM. Determination of autoantibodies for AITD and celiac disease should be performed at the diagnosis of diabetes and at least annually [47]. Testing for anti-21-hydroxylase autoantibodies should be performed as well in patients with T1DM and AITD since a positive screening is highly predictive of future adrenal failure.

Patients with a diagnosis of APS-2 and their first-degree relatives should be periodically monitored in the same way [40]. Autoantibodies can develop at any age and are frequently detectable years before disease onset. Although the optimal screening interval is not defined, repeated testing is mandatory. Once autoimmunity is detected, the following functional tests are needed to confirm the diagnosis [41].

#### 2.2.3. Clinical Features

The pathophysiology, clinical signs and symptoms, and laboratory findings of AD in APS-2 are identical to those previously described. The exception is that APS-2 mainly affects women, three times more than men, between 30 and 40 years of age [11]. AD may cause a decrease in insulin need with hypoglycemia, as well as a thyrotropin (TSH) increase due to the absence of glucocorticoid-transmitted inhibition of TSH secretion [48]. In general, patients with AD experience impaired quality of life (QoL), even though outside the APS context. Lower QoL outcomes have been shown to be associated with female sex, higher age, and autoimmune comorbidities [49]. Moreover, QoL impairments are particularly pronounced for patients with concomitant T1DM [50]. Useful tools for evaluating QoL in AD patients, such as the recently developed disease-specific questionnaire AddiQoL [51], might therefore be used to detect minor changes in well-being during routine follow-ups and in future studies.

The clinical presentation and diagnosis of AITD and other component disorders of APS-2 are the same as that of individual diseases. AITD occurs in 70% of patients with APS-2, representing the most frequent autoimmune endocrinopathy, in contrast with APS-1, in which AITD is a minor component (8–40%) [21,22]. The autoimmune etiology of primary hypothyroidism (elevated serum TSH and low/normal serum thyroid hormone levels) or hyperthyroidism (low serum TSH with elevated/normal serum thyroid hormone levels) is confirmed by the presence of anti-TPO, anti-Tg, and anti-TSH receptor antibodies [52]. 

T1DM is a common component disorder of APS-2 (40–60%) and is often its first manifestation, and the combination of T1DM with AITD is the most prevalent phenotype in APS-2 [41]. Although T1DM was once considered to be exclusively a pediatric disease, it can occur at any age, with a peak in incidence around puberty [53]. Patients with T1DM show a high frequency of positive anti-GAD-65, IA2, and ZnT8 antibodies. The precise immunopathogenesis in the context of APS-2 has not been fully elucidated, although evidence for shared immunologic mechanisms induced by environmental factors in a genetic background is present, as abovementioned. Consideration should therefore be given to the manifestation of T1DM within the larger context of APS-2, in order to screen for additional autoimmune glandular and non-glandular diseases serologically and functionally. The concomitant presence of AD and T1DM leads to hypoglycemic episodes due to reduced gluconeogenesis and greater insulin sensitivity; clinicians should thus rule out a potential AD onset in patients with T1DM suffering from unexplained recurrent hypoglycemia and fatigue [54]. Hypothyroidism in a patient with T1DM may be accompanied by hypoglycemia due to reduced insulin requirements and increased insulin sensitivity, whereas hyperthyroidism is associated with hyperglycemia and higher insulin need [11].

Affected patients may develop additional autoimmune conditions including primary hypogonadism, vitiligo, chronic atrophic gastritis (with or without pernicious anemia), autoimmune hepatitis, celiac disease, stiff-man syndrome (a rare neurological disease characterized by progressive rigidity and spasms), myasthenia gravis, and alopecia [3,40,55].

#### 2.2.4. Treatment

Treatment of APS-2 depends upon which organs are affected, focusing on the identification and management of the underlying autoimmune disorders. A multidisciplinary form of management is mandatory. Improper follow-up may put the patient at risk of adrenal crisis, hypoglycemia, and diabetic ketoacidosis, among others. Family members must be carefully followed up, as well [56].

AD is treated by replacing the missing hormones, cortisol and aldosterone, with glucocorticoids and mineralocorticoids (as described before). Thyroid replacement therapy in hypothyroidism proceeds with levothyroxine. In a patient with coexisting hypothyroidism and AD, starting levothyroxine prior to the replacement of glucocorticoids can precipitate acute adrenal insufficiency and crisis [38]. Anti-thyroid drugs, radioactive iodine treatments, or thyroidectomy are used to treat hyperthyroidism due to the autoimmune Graves’ disease. Lifelong insulin is the mainstay treatment for T1DM, aiming to maintain glucose control as near to normal as safely possible. The advanced hybrid closed-loop (HCL) systems are currently available in pediatric patients with T1DM. These systems integrate insulin infusion with continuous glucose monitoring (CGM), automatically increasing, decreasing, and suspending basal insulin delivery in response to CGM [57,58]. The HCL system has improved glucose control, disease management, and QoL in children and young people with T1DM and their caregivers [59]. Recent advances in the knowledge of the immune pathogenesis of T1DM have also led to innovative approaches, such as immunotherapies targeting T-cells [60], intramuscular injections of vitamin B12, a gluten-free diet, and other treatments, including immunosuppressant medications if needed, depending on the possible occurrence of less-common manifestations [41].

### 2.3. IPEX Syndrome

The classical triad of symptoms displayed by more than half of IPEX patients is enteropathy, T1DM, and dermatitis. Autoimmune manifestations involving other target organs may develop, leading to a certain overlap with APS-1, although they usually develop much earlier in life in IPEX syndrome [40].

#### 2.3.1. Epidemiology

It is an extremely rare hemizygous disorder with a prevalence below 1:1,000,000, typically inherited in boys in an X-linked recessive manner, and might occur perinatally or later in life, but the onset is most common within the first few weeks or months of life [6]. IPEX can be fatal if not diagnosed and treated early with immunosuppressive agents or, if possible, hematopoietic stem cell transplantation (HSCT).

#### 2.3.2. Pathogenesis and Autoimmunity

IPEX syndrome is caused by mutations of the *FOXP3* gene located in the short arm zone (Xp11.23) of the X chromosome. *FOXP3* encodes a forkhead-winged helix transcription factor (FOXP3, also known as scurfin) primarily expressed by CD4^+^CD25^+^ regulatory T-cells (Tregs) and indispensable for their function and maintenance [61]. The inability of FOXP3 to bind DNA in Tregs impairs their immune suppressor function, preventing them from inhibiting proliferation and cytokine secretion of effector T-cells and leading to massive autoimmunity [3]. Despite being considered a Tregs-dysfunction immune disease, in IPEX syndrome, B-cells can be both direct and indirect targets of Tregs-mediated suppression [62]. FOXP3 deficiency with the consequent lack of Tregs is in fact associated with abnormal B-cell development, loss of B-cell anergy, and accumulation of autoreactive clones in the periphery, due to the disrupted control of peripheral B-cell tolerance checkpoints, causing the production of tissue-specific autoantibodies, partly responsible for the clinical manifestations. Autoantibodies to enterocyte antigens, harmonin and villin in particular, might be used as specific and sensitive markers of IPEX syndrome enteropathy, proving to be useful for screening and clinical monitoring of affected children [63]. If T1DM and AITD are present, specific autoantibodies may be observed [6].

An increasing number of syndromes sharing the clinical manifestations of IPEX in the presence of a wild-type *FOXP3* gene have recently been described. The underlying defect is generally unknown in most cases, although IPEX-like phenotypes have been associated with various novel genetic syndromes [6]. Gambineri et al. provided a comprehensive analysis of the clinical features and molecular bases of a cohort of patients with IPEX phenotypes. They demonstrated that about half of patients had no identifiable *FOXP3* mutations but carried a disease-associated variant affecting a distinct gene (*LRBA*, *STAT1*, *STAT3*, *CTLA4*, *IL2RA*, *STAT5B*, *DOCK8*, *TTC37*, and *TTC7A*) [64]. For confirmation of the disease type, the genomic testing (next generation sequencing, whole-exome sequencing) is therefore highly recommended.

#### 2.3.3. Clinical Features

Enteropathy is the most common and representative feature of IPEX syndrome, as well as usually the first manifestation. It is characterized by diarrhea (intractable in some cases), hematochezia, and vomiting, affecting 97.9% of patients [65]. Intestinal involvement results in malabsorption syndrome, weight loss, and failure to thrive. Since villin and harmonin are also distributed in the proximal renal tubules, some patients experience nephropathy, and the protracted exposure to nephrotoxic drugs may contribute [6]. 

Endocrinopathies include T1DM and AITD. Diabetes may develop as early as 2 days of age, representing the second hallmark of this disorder. AITD can present in the form of autoimmune thyroiditis with hypothyroidism or hyperthyroidism [21].

Cutaneous lesions may present as eczematiform (mainly atopic dermatitis), ichthyosiform, and psoriasiform, or combinations of them [66]. The presence of atopic dermatitis correlates with high levels of serum IgE and peripheral blood eosinophilia, as they represent an early hallmark of the disease, helping in the process of differential diagnosis [67].

Besides the significant triad of enteropathy, endocrinopathy, and skin-related disorders, other symptoms may complicate the clinical picture. Additional disorders include autoimmune hematologic disorders (hemolytic anemia, thrombocytopenia, neutropenia), autoimmune hepatitis, and arthritis involving one or more joints [68,69]. In children, splenomegaly and lymphadenopathy as consequences of an ongoing excessive autoimmune lymphoproliferation might be observed. The clinical spectrum can be worsened by infections, in particular invasive bacterial infections (sepsis, meningitis, pneumonia, osteomyelitis). Factors that contribute to the susceptibility to infections of IPEX patients include, among others, immune suppressive therapy, autoimmune neutropenia, and impaired intestinal and cutaneous permeability barrier function [68].

Laboratory features reflect the alterations of the nutritional status and glucose metabolism, related to severe enteropathy and T1DM. Elevated serum IgE levels and increased eosinophil counts are typical, with normal levels of IgM, IgG, and IgA, unless a wasting syndrome is present. Mutational analysis of the *FOXP3* gene should be performed in children with the classic triad of symptoms and elevated IgE, especially if a reduced number of Tregs is found [68]. Most of the patients are positive for a variety of autoantibodies, including anti-harmonin and anti-villin autoantibodies (useful for screening and clinical monitoring), as well as T1DM- and AITD-associated autoantibodies [63,68,70].

#### 2.3.4. Treatment

If untreated, infants with IPEX fail to thrive and are likely to die in early childhood [3,21]. Multidisciplinary supportive care usually includes nutritional treatment (fluid resuscitation, total parenteral nutrition), insulin (T1DM might be extremely difficult to manage due to intractable diarrhea), levothyroxine replacement, antimicrobials, albumin, and blood products. Additional specific care and therapies are required if other autoimmune disorders develop. Inhibition of T-cell activation by glucocorticoids or steroid-sparing agents (cyclosporine, tacrolimus, sirolimus) is the treatment of choice for the chronic management of IPEX patients. Allogeneic HSCT still remains the only curative therapy available [6].

An international multicenter retrospective study that analyzed 96 patients demonstrated similar overall survival regardless of whether they had received immunosuppression (86.8% at 15 years) or HSCT (73.2% at 15 years). The latter had lower, although not significantly, survival rates attributable to the high mortality rate within the first years after transplantation. The HSCT group showed instead higher rates of disease-free survival, thanks to the stable resolution of autoimmunity as opposed to the persistent disease progression in the non-HSCT group [71].

Innovative gene-therapy-based approaches are currently being studied for clinical application for patients with IPEX [72].

## 3. Conclusions

APS-2 is the most common among polyglandular syndromes, particularly if considered in its version described by the most recent nosological classification. Nevertheless, APS-1 is certainly the most peculiar form found in the pediatric age. In all cases of children suffering from an autoimmune disorder (e.g., T1DM, AITD, AD), periodic clinical and laboratory screening must be performed in search of any other autoimmune manifestations, in such a way as to intercept them early and intervene promptly with adequate treatment, optimally in an interprofessional setting where outcomes are highly improved. Children presenting with the classic symptom triad of IPEX syndrome should be recognized and treated immediately. Further studies will be necessary in order to develop and implement innovative gene therapies, able to improve the prognosis of this disease, which is dramatically poor in the first years of life when untreated.

## Figures and Tables

**Figure 1 children-10-00588-f001:**
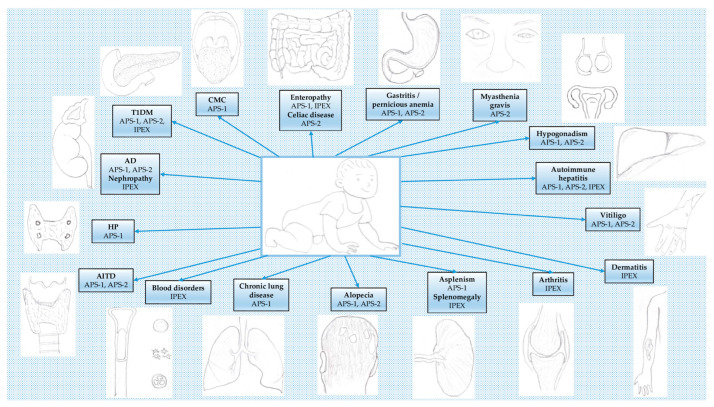
Main manifestations of autoimmune polyendocrine syndromes. Abbreviations: APS, autoimmune polyglandular syndrome; IPEX, immune dysregulation, polyendocrinopathy, enteropathy, X-linked; CMC, chronic mucocutaneous candidiasis; HP, hypoparathyroidism; AD, Addison disease; T1DM, type 1 diabetes mellitus; AITD, autoimmune thyroid disease.

**Table 1 children-10-00588-t001:** Features of autoimmune polyglandular syndromes.

	APS-1	APS-2	IPEX
Incidence	<1:100,000/year	1-2:100,000/year	<1:1,000,000 (prevalence)
Age of onset	Infancy/early childhood	Late childhood/early adulthood	Perinatally/within the first few weeks or months of life
Gene and inheritance	Monogenic (*AIRE*, chromosome 21q22.3), autosomal recessive	Polygenic, associated with HLA and non-HLA genes	Monogenic (*FOXP3*, chromosome Xp11.23), X-linked recessive
Autoantibodies	Anti-interferon-ω/α2	Organ-specific	Anti-harmonin, anti-villin
Common phenotype	Candidiasis, hypoparathyroidism, Addison disease	Addison disease, type 1 diabetes mellitus, autoimmune thyroid disease	Enteropathy, type 1 diabetes mellitus, dermatitis
Main treatments	Antifungal therapy, calcium and vitamin D, hydrocortisone and fludrocortisone	Hydrocortisone and fludrocortisone, insulin, levothyroxine	Nutritional treatment, insulin, levothyroxine, immunosuppressive agents, HSCT

Abbreviations: APS, autoimmune polyglandular syndrome; *AIRE*, autoimmune regulator; *FOXP3*, forkhead box protein P3; HLA, human leukocyte antigen; HSCT, hematopoietic stem cell transplantation.

## Data Availability

Data sharing not applicable.

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
