# Peer review of "Autoimmune Polyendocrine Syndromes in the Pediatric Age"

_children, 2023, doi:10.3390/children10030588_

Round 1
Reviewer 1 Report
1. Du we know anything about QoL in patients with APS1/2 compared to "regular" autoimmune thyroiditis/Addison?
2. You write about immunotherapy as treatment of type 1 diabetetes. A huge advance in the insulin treatment are the HCL pumps which you could write about
Author Response
Reviewer 1
We do thank the reviewers for the careful reading of the manuscript and the constructive remarks. We believe they have helped us to substantially improve our manuscript. We provided detailed point-by-point responses to all comments in order to improve and clarify the manuscript. All changes in the revised manuscript are highlighted in light yellow (rev 1) or light green (rev 2). Reviewers’ original comments are listed below (in italics) followed by our responses to each comment.
- Do we know anything about QoL in patients with APS1/2 compared to "regular" autoimmune thyroiditis/Addison?
We think this is an excellent comment, thank you for pointing this out. Since autoimmune polyendocrine syndromes are very rare disorders in the pediatric age, strong data about the quality of life (QoL) in these patients is lacking. Anyway, we mentioned in the text what is currently known about reduced QoL in Addison’s disease, especially if an autoimmune etiology and autoimmune comorbidities are present.
- You write about immunotherapy as a treatment of type 1 diabetes. A huge advance in the insulin treatment are the HCL pumps which you could write about.
Thank you for the reminder. As suggested, we included a more detailed description of insulin treatment with HCL pumps in the revised version of the manuscript. We also mentioned the improved diabetes management and QoL thanks to HCL systems.
Reviewer 2 Report
To the authors
The paper is well written and updated.
However, in some sections the authors simply present the clinical features of the 3 main diseases (that are well known), without providing a more critical approach. Therefore, as presented, it does not add much to already published literature.
I have some suggestions to improve the readability of this paper:
-please refer to the most recent IUIS classification of inborn errors of immunity (specifically, APS1 and IPEX are classified into immune dysregulation disorders)
-IPEX: recently, many diseases with IPEX-like phenotype have been described (i.e CTLA4 and LRBA deficiency, STAT3 gain of function, and others). Please consider to discuss it in this section (see Gambineri et al, IPEX-like diseases).
-Diagnostic approach: as correctly stated, the approach could be complicated by the presence of extra-endocrinological features. I suggest to include a brief discussion on how to approach a patient with multiple endocrine autoimmunity (.ie if the patient has T1diabetes and thyroiditis what wuold you suggest? Or: when you have a patient with early onset endocrinological autoimmunity, do you suggest to routinely screen the patient for other autoimmune endocrinopathies?). I think that this will integrate the descriptive sections of this paper and make it more attractive.
-Different items could help in the process of differential diagnosis: i.e eosinophilia in IPEX and IPEX-like disorders, features deriving from an extended immunological assessment... Also, the role of morbidity (especially in this last category of diseases) and extra-endocrinological features (autoimmunity, lymphoproilferation) should be clearly presented.
Minor comment: L 289 please refer to HSCT and not to bone marow transplantation
Author Response
Reviewer 2
We do thank the reviewers for the careful reading of the manuscript and the constructive remarks. We believe they have helped us to substantially improve our manuscript. We provided detailed point-by-point responses to all comments in order to improve and clarify the manuscript. All changes in the revised manuscript are highlighted in light yellow (rev 1) or light green (rev 2). Reviewers’ original comments are listed below (in italics) followed by our responses to each comment.
To the authors
The paper is well written and updated.
However, in some sections the authors simply present the clinical features of the 3 main diseases (that are well known), without providing a more critical approach. Therefore, as presented, it does not add much to already published literature. I have some suggestions to improve the readability of this paper:
-please refer to the most recent IUIS classification of inborn errors of immunity (specifically, APS1 and IPEX are classified into immune dysregulation disorders)
Thank you for the reminder. As suggested, we mentioned the IUIS classification and added it to the Introduction.
-IPEX: recently, many diseases with IPEX-like phenotype have been described (i.e. CTLA4 and LRBA deficiency, STAT3 gain of function, and others). Please consider to discuss it in this section (see Gambineri et al, IPEX-like diseases).
As suggested, we discussed IPEX-like disorders in the “IPEX - Pathogenesis and autoimmunity” section. Thank you for the reference suggestion.
-Diagnostic approach: as correctly stated, the approach could be complicated by the presence of extra-endocrinological features. I suggest to include a brief discussion on how to approach a patient with multiple endocrine autoimmunity (i.e. if the patient has T1diabetes and thyroiditis what would you suggest? Or: when you have a patient with early onset endocrinological autoimmunity, do you suggest to routinely screen the patient for other autoimmune endocrinopathies?). I think that this will integrate the descriptive sections of this paper and make it more attractive.
Thank you for pointing this out, we agree that this is an excellent consideration, since screening is a fundamental aspect in the early diagnosis and management of APSs. They usually begin with a single endocrinopathy, and the further involvement of other endocrine and/or non-endocrine organs may take up to years or decades. Therefore, high clinical suspicion should be maintained for the presence of other autoimmune diseases. In fact, we already indicated that:
- “any child with refractory CMC should be evaluated for coexisting endocrine disorders” (APS-1 – clinical feature)
- “children with a mono glandular autoimmune disease should undergo clinical and serological screening in order to intercept other glandular and/or non-endocrine autoimmune disorders. Patients with a diagnosis of APS-2 and their first-degree relatives should be periodically monitored in the same way” (APS-2 – Pathogenesis and autoimmunity)
- subjects with AD should be screened for other autoantibodies (for AITD, T1DM, celiac disease, or other autoimmune endocrinopathies and extra-endocrinological disorders) (APS-2 – Pathogenesis and autoimmunity)
- “although the optimal screening interval is not defined, repeated testing is mandatory. Once autoimmunity is detected, the following functional tests are needed to confirm the diagnosis” (APS-2 – Pathogenesis and autoimmunity)
- “Consideration should therefore be given to the manifestation of T1DM within the larger context of APS-2, in order to screen for additional autoimmune glandular and non-glandular diseases serologically and functionally” (APS-2 – Clinical features)
- “clinicians should thus rule out a potential AD onset in patients with T1DM suffering from unexplained recurrent hypoglycemia and fatigue” (APS-2 – Clinical features)
- “Mutational analysis of the FOXP3 gene should be performed in children with the classic triad of symptoms and elevated IgE, especially if a reduced number of Tregs is found” (IPEX – Clinical feature)
- “In all cases of children suffering from an autoimmune disorder (e.g., T1DM, AITD, AD), periodic clinical and laboratory screening must be performed in search of any other autoimmune manifestation” (Conclusions)
It should also be considered that:
- every center has a different approach to patients with single or multiple endocrine autoimmunities,
- there is a great clinical variability with inter-individual differences in APSs presentation,
- the purpose of our Review is mainly to summarize the features of APSs in children, given the rarity and the complexity of these disorders in the pediatric age.
Following your suggestion, in the revised version of the manuscript, we thought it appropriate to strengthen the importance of further screening in a patient diagnosed with a single autoimmune endocrine disorder, as well as highlight that these individuals should be rescreened for autoantibodies for other autoimmune conditions at appropriate intervals even if their initial autoantibody tests are negative (see 2.1.3. and 2.2.2. sections).
-Different items could help in the process of differential diagnosis: i.e., eosinophilia in IPEX and IPEX-like disorders, features deriving from an extended immunological assessment... Also, the role of morbidity (especially in this last category of diseases) and extra-endocrinological features (autoimmunity, lymphoproliferation) should be clearly presented.
We thank the reviewer for the comment. We were glad to broaden the description of morbidities and additional features in the revised version of the manuscript as suggested (IPEX – Clinical features).
Minor comment: L 289 please refer to HSCT and not to bone marrow transplantation
Thank you. The correction has been made.